

# Seasonal variability in evapotranspiration partitioning and its
# relationship with crop development and water use efficiency
# of winter wheat
Ying Ma[1], Praveen Kumar[2], and Xianfang Song[1]
[1]Key Laboratory of Water Cycle and Related Land Surface Processes, Institute of Geographic
Sciences and Natural Resources Research, Chinese Academy of Sciences, Beijing 100101,
China
[2]Department of Civil and Environmental Engineering, University of Illinois at
Urbana-Champaign, Urbana 61801, IL, USA
*Corresponding to:* Ying Ma (maying@igsnrr.ac.cn)

## 15 Abstract

The partitioning of evapotranspiration (ET) into soil evaporation (E) and crop transpiration (T)
is fundamental for accurately monitoring agro-hydrological processes, assessing crop
productivity, and optimizing water management practices. In this study, the isotope tracing
technique was used to partition ET and quantify the root water uptake sources of winter
wheat during the 2014 and 2015 growing seasons in Beijing, China. The correlations between
seasonal ET partitioning and the leaf area index (LAI), grain yield, and water use efficiency
(WUE) were investigated. The fraction of T in ET ($F_T$) between the greening and harvest





seasons was 0.82 on average and did not vary significantly among the different irrigation and
fertilization treatments ($p > 0.05$). However, the values of $F_T$ during the individual growth
periods were remarkably distinct (ranging from 0.51 to 0.98) among the treatments. The
seasonal variability in $F_T$ could be effectively explained via a power-law function of the LAI
($F_T = 0.61 \, \text{LAI}^{0.21}$, $R^2 = 0.66$, $p < 0.01$). There was no significant relationship between $F_T$ and
the grain yield or WUE ($p > 0.05$). The total T during the jointing-heading and heading-filling
periods ($T_{jf}$) had significantly quadratic relationships with the crop yield and WUE ($p < 0.01$).
Both the crop yield and the WUE had high values under the $T_{jf}$ range of 117.5-155.8 mm.
Furthermore, the WUE was improved by increasing the ratio of E in ET ($F_E$) during the
greening-jointing period and by reducing $F_E$ during the filling-harvest period. Winter wheat
mainly utilized soil water from the 0-20 cm (67.0%), 20-70 cm (42.0%), 0-20 cm (38.7%),
and 20-70 cm (34.9%) layers during the greening-jointing, jointing-heading, heading-filling,
and filling-harvest periods, respectively. This indicated that the irrigation wetting layer
should be controlled at depth of 70 cm to conserve water.
**1   Introduction**
Evapotranspiration (ET) represents a critical component of the water cycle in the
soil-plant-atmosphere continuum (SPAC), which is fundamental for crop development and
for determining water use efficiency (WUE). Partitioning ET into soil evaporation (E) and
plant transpiration (T) can provide deep insight into an evaluation of the water saving
potential and optimization of agro-management practices (Newman et al., 2006; Guan and
Wilson, 2009; Agam et al., 2012). The majority of previous studies referred to T as the



productive component of the crop yield, while E was described as the non-productive water
loss (Agam et al., 2012; Van Halsema and Vincent, 2012; Ding et al., 2017). A positive linear
function has generally been used to describe the relationship between the yield and T
provided that water was the main limiting factor on the yield (Hanks, 1983; Ben-Gal and
Shani, 2002; Tolk and Howell, 2009). Nevertheless, some studies claimed that E might
indirectly benefit the crop yield by creating a microclimate that is more favorable for crop
growth and productivity (Stanhill, 1973; Tolk et al., 1995; Burt et al., 2005; Kustas and Agam,
2013). Therefore, further research is necessary to separate the E and T components of ET and
investigate their interrelationships with crop development and the WUE.
The partitioning of ET has been studied using several methods and techniques (Kool et al.,
2014; Sutanto et al., 2014; Sprenger et al., 2016). Conventional hydrometric methods
employed various techniques to directly measure ET (e.g., the eddy covariance technique,
micro Bowen ratio energy balance method, and the weighing lysimeter approach) in addition
to E (via a micro-lysimeter) or T (using sap flow sensors) (Mitchell et al., 2009; Cavanaugh
et al., 2011; Liu et al., 2002; Zhang et al., 2003; Sun et al., 2006). Simulation models such as
the Shuttleworth-Wallace model, the Food and Agriculture Organization (FAO) dual crop
coefficient model, the HYDRUS-1D model, and the hybrid dual-source model (TVET) have
also been used to simultaneously calculate E and T (Li et al., 2010; Zhang et al., 2013; Ding
et al., 2017; Sutanto et al, 2012; Guan and Wilson, 2009). Since various fractionation
processes such as condensation and evaporation leave characteristic imprint on the isotopic
composition of water, the stable water isotopes of $^{18}$O and D are considered ideal (natural)
tracers for separating E and T from ET and tracking water through the soil based on distinct



isotopic signatures of water fluxes (Brunel et al., 1997; Sprenger et al., 2016). Water isotopes
are highly fractionated during E processes, causing the remaining soil water to become
enriched in heavy isotopes; meanwhile, T does not modify the isotopic composition, since
there is typically no isotopic fractionation during water uptake or transport through roots and
stems (Wang et al., 2010a; Sutanto et al, 2014).

Due to recent advances in the techniques and instruments used to collect measurements of

$\delta^{18}O$ and $\delta D$ for both liquid water and water vapor, isotope-based methods have been
increasingly applied to agricultural systems to precisely partition ET at different time scales
(Wang et al., 2012; Wang and Yamanaka, 2014; Zhang et al., 2011; Wang et al., 2016; Lu et
al., 2017; Wei et al., 2018). It was reported that T accounts for 20-80% of the total seasonal
ET in sparse canopies and row crops, especially under arid and semi-arid conditions (Agam
et al., 2012; Coenders-Gerrits et al., 2014; Kool et al., 2014; Lu et al., 2017). At the daily
scale, the ratio of T within ET ($F_T$) varied over a wide range of 0.2-1 within a rice paddy field
during a complete growing season in Mase, Tsukuba (Wei et al., 2015). The daily $F_T$ also
changed greatly (0.52-0.96) throughout the growing season of maize in northwestern China
(Wen et al., 2016; Wu et al., 2017). Substantial differences in $F_T$ were discovered between
the late filling stage (0.83) and the stage of wax ripeness (0.6) in an irrigated field of winter
wheat in the North China Plain (NCP) (Zhang et al., 2011). The values of $F_T$ changed from
0.46 to 0.74 after an irrigation event during the early growth stage of winter wheat in the
NCP and in central Morocco (Wang et al., 2012; Aouade et al., 2016). These studies revealed
very distinct changes in $F_T$ throughout the crop-growing season and the significant influence



of irrigation on the partitioning of ET. It is therefore necessary to thoroughly clarify the
seasonal variability in the partitioning of ET in association with its major influencing factors.
The seasonal variations in ET partitioning are strongly associated with crop development
(Sprenger et al., 2016). The leaf area index (LAI) is often regarded as an effective crop
parameter for explaining the variabilities in the E/ET ratio ($F_E$) and $F_T$. It is commonly
believed that $F_E$ decreases exponentially with the LAI for most crops and that the $F_T$
increases logarithmically with an increase in the LAI in the absence of water stresses
(Villalobos and Fereres, 1990; Liu et al., 2002; Yu et al., 2009; Kato et al., 2004). However,
Kang et al. (2003) proposed that $F_T$ and the LAI exhibited a saturation relationship for wheat
and maize in a semi-arid region of Northwest China. Several recent studies identified a
power-law correlation between $F_T$ and the LAI for agricultural systems at both the global
scale and in certain croplands (Wang et al., 2014; Wei et al., 2015; Wu et al., 2017; Lu et al.,
2017; Zhao et al., 2018). In addition, numerous possibilities were suggested for high $F_T$ even
under low LAI conditions. To illustrate the global variability in the partitioning of ET, Wang
et al. (2014) further developed a function relating $F_T$ to the growth stage relative to the timing
of the peak LAI. It was evident that the LAI within different growth stages should be utilized
to evaluate the variability in ET partitioning and crop water use capabilities.
The ability of a crop to access water resources from different soil horizons can be
estimated via the root water uptake (Asbjornsen et al., 2007; Wang et al., 2010b; Zhang et al.,
2011; Yang et al., 2015). Common methods applied to assess water uptake patterns include
the IsoSource model in addition to less than three-layer linear mixing models and Bayesian
mixing models (Phillips and Gregg, 2003; McCole and Stern, 2007; Moore and Semmens,



2008; Stock and Semmens, 2013). The MixSIAR framework is the latest Bayesian stable
isotope analysis mixing model in R that considers multiple sources of uncertainty and
provides definite proportions of source contributions. It has been employed successfully to
determine the contributions of soil water at different layers to the water uptake of summer
maize (Ma and Song, 2016). The root water uptake also indicates the availability of soil water
resources to crops, and it varies with different agricultural management practices. Therefore,
combining the seasonal partitioning of ET with the development of the LAI and root water
uptake patterns can provide a comprehensive understanding of E and T processes. It also help
design a reasonable irrigation depth, which is vital for improving the crop yield and WUE in
regions with a high food demand and limited water resources such as in the NCP.

The NCP constitutes one of the major food production regions in which winter wheat

represents the main water-consuming crop. In addition, the NCP provides approximately 69%
of the wheat production for all of China. However, irrigated agriculture practices throughout
the NCP are facing critical challenges (i.e., very limited water supplies) to the provision of
sufficient quantities of food. To optimize the irrigation strategies for winter wheat,
considerable research has been conducted to determine the relationships among the seasonal
ET with the crop yield and WUE (Li et al., 2005; Sun et al., 2006; Shang et al., 2006; Liu et
al., 2013). E and T were also partitioned, and an exponential relationship between $F_E$ and the
LAI was established (Liu et al., 2002; Yu et al., 2009). Furthermore, E was reported to be an
unproductive water loss, and thus, it should be reduced in regions with a severe water deficit.
Recently, Zhang et al. (2011) simultaneously addressed the partitioning of ET and the
characterization of root water uptake depths for winter wheat during the growing season.



However, the definite correlations between the magnitude and fraction of seasonal ET
partitioning with the grain yield and WUE are still unclear. Further investigations are
therefore required to demonstrate seasonal variations of ET partitioning and root water uptake
pattern and quantify their relationships with the LAI, grain yield and WUE under different
agricultural management practices.

In this study, the isotope mass balance approach was utilized in conjunction with the soil

water balance method to partition the ET of winter wheat during the 2014 and 2015 growing
seasons in Beijing, China. The three primary objectives of this study were to (1) detect the
variations of ET partitioning during the different growth stages of winter wheat, (2) quantify
the seasonal root water uptake patterns of winter wheat, and (3) determine the relationships
between ET partitioning and the LAI, grain yield and WUE. The results were applied to
establish optimal agricultural management practices and design the irrigation depth.
**2   Materials and methods**
**2.1   Field experiments**
Field experiments with winter wheat were conducted from 2013 to 2015 at the Irrigation
Experiment Station of the China Institute of Water Resources and Hydropower Research
(IWHR) at Daxing, Beijing (39°37′N latitude, 116°26′E longitude, 40.1 m a.s.l. elevation).
The climate in this area is sub-humid with a mean annual precipitation of 540 mm, but only
20-30% of the precipitation occurs during the winter wheat season (Cai et al., 2009). The
mean annual temperature is 12.1 ℃ and mean seasonal reference evapotranspiration ($ET_0$) of
winter wheat is 610 mm. Soils in the 2-m profile were sampled every 20 cm depth to measure
their physical and chemical properties. The depths with similar soil particle size and organic



carbon were merged into one layer. The soil profile was finally divided into four layers and
their main properties are shown in Table 1.

The winter wheat (variety: Zhongmai-175) was planted on October 9, 2013, and

harvested on June 8, 2014, during the 2014 growing season. The sowing and harvest dates
during the 2015 growing season were October 11, 2014, and June 8, 2015, respectively. Five
irrigation and fertilization treatments (T1, T2, T3, T4, and T5) were applied from winter
greening to the harvest season. Here, treatment T5 refers to the agricultural management
practices employed by local farmers with a total irrigation of 240 mm and a nitrogen supply
of 210 kg N ha$^{-1}$ (as urea) (Table 2). In comparison with the reference treatment (T5), the T1
and T2 treatments had reduced irrigation of 60 mm from greening to jointing and 80 mm
from the filling to the harvest period, respectively. The T3 and T4 treatments both reduced the
irrigation during jointing-heading or heading-filling stage by 80 mm compared with treatment
T5. The nitrogen (N) application rates for the T1 and T3 treatments were both 0.5-fold of that
for treatment T5, while 1.5-fold of the nitrogen in treatment T5 was applied to both the T2
treatment and the T4 treatment. All the irrigation was provided in a single application per
stage. The application date was 27 Mar, 22 Apr, and 22 May in 2014, while it was 29 Mar, 9
May, and 21 May in 2015, respectively. The detailed irrigation and fertilization schedules for
these five treatments are shown in Table 2. Three replicates were conducted for every
treatment in the plots with each area of 6 ×5 m. Basin irrigation with groundwater was
implemented for all of the treatments. Precision-leveled basins were used to prevent runoff.

The soil water contents in the 2-m soil profile were measured at a 20-cm interval every

5-7 days in each plot using a TRIME-IPH probe based on the Time domain Reflectometry





with Intelligent MicroElements technique (IMKO GmbH, Ettlingen, German). Additional
measurements were conducted when soil water samples were collected for isotope analysis as
well as before and after each irrigation or heavy rainfall event. Meanwhile, three plants in
each plot were selected to manually observe their leaf areas (obtained by multiplying the leaf
length and width), which were then calibrated using a leaf scanner (F915900, Canon, Canada).
The LAI was calculated as the product of the calibrated leaf area per plant and the number of
plants per unit area. The grain was air-dried, and the crop yield was recorded separately for
each plot after harvesting.

Meteorological data including the precipitation, maximum and minimum air temperatures,

solar radiation, wind speed and relative humidity were recorded every 30 min by the
automatic weather station (Monitor Sensors, Caboolture QLD, Australia). The rainfall
amounts were 77.0 mm and 74.7 mm between the greening and harvest seasons in 2014 and
2015, respectively. Both seasons were dry at 75% precipitation exceedance probabilities
(PEPs) in terms of the rainfall frequencies during the last five decades in the Beijing area.
However, there was an additional 34.8 mm of precipitation during the greening-jointing
period and 26.9 mm less precipitation during the jointing-heading period in 2015 relative to

2014.

**2.2   Water sampling and isotopic analyses**
Different waters including the precipitation, irrigation water, soil water, and stem water were
sampled to analyze the isotopic composition of $^{18}$O and D. The precipitation was collected
after each rainfall event via a rain collector coupled with a polyethylene bottle and funnel. A
ping-pong ball was positioned at the funnel mouth (Wang et al., 2012). The ping-pong ball





floated up when rain fell at the funnel mouth and enabled the rainfall to move into the bottle.
Evaporation was then prevented during the rainfall process. The collected rainwater was
transferred to a bottle immediately, sealed and stored. Irrigation water was sampled in each
irrigation event.

Three stems of each treatment were sampled at an interval of about one week. Each stem

was taken from the part between the soil surface and the first node of one representative plant.
It was cut into pieces in 2-3 cm length, then put into a vial and sealed with parafilm. All the
epidermises of the stems were removed to eliminate the effect of the isotopically depleted
atmospheric water vapor on the stem water isotopic compositions (Brunel, 1997).

The soil water at depths of 10, 20, 30, 50, 70, 90, 110, 150, and 200 cm was sampled on

and after the day of collecting stem water, and after each irrigation or heavy rainfall event. A
suction lysimeter made of a Teflon pipe and porous ceramic cup was installed and used to
abstract the soil water at each depth (Wang et al., 2012). If the soil water content was too low
to collect soil water by the suction lysimeter, soil sample instead was collected using a hand
auger.

All of the stem and soil samples were kept refrigerated (-15 ℃ to -20 ℃) prior to

measuring the isotopic compositions. The cryogenic vacuum distillation system (LI-2000,
LICA, Beijing, China) was applied to extract water in the soil and stem samples (West et al.,
2006). The ratio of $^2H/^1H$ and $^{18}O/^{16}O$ of different water samples were measured on a Los
Gatos Research (LGR) DLT-100 liquid water isotope analyzer (San Jose, CA, America). They
were calibrated against the VSMOW international standards and converted to $\delta D$ and $\delta^{18}O$
values. The measuring precision for $\delta D$ and $\delta^{18}O$ was ±1‰ and ±0.1‰, respectively.



### 2.3 Evapotranspiration partitioning methods

Transpiration changes soil water content but keep soil water isotopic composition constant

because water uptake from soil by plants does not result in isotopic fractionation

(Zimmerman et al., 1967). On the contrary, both soil water content and soil water isotopic

composition are changed in evaporation process (Allison and Barnes, 1983). Many previous

studies reported that the water balance and isotope mass balance equations were robust to

partition ET into E and T when sampling intervals were short (Hsieh et al., 1998; Robertson

and Gazis, 2006; Wenninger et al., 2010; Wang et al., 2012). In this study, the ET in the day

of the stem water sampling was partitioned into E and T using the following soil water

balance and isotope mass balance equations in the 0-200 cm profile:

$$m_f - m_i = m_P + m_I - m_{ET} - m_D - m_R \tag{1}$$

$$m_{ET} = m_E + m_T \tag{2}$$

$$\delta_E m_E + \delta_T m_T = \delta_i m_i + \delta_P m_P + \delta_I m_I - \delta_f m_f - \delta_D m_D - \delta_R m_R \tag{3}$$

where $m$ and $\delta$ represent the water flux and isotopic composition of $\delta^{18}O$ in different waters,

respectively, $f$ and $i$ denote the final and initial state of the soil water storage in one sampling

day of stem water, respectively, P is the precipitation, I is the irrigation, D is the drainage out

of the soil profile, and R is the surface runoff. There were two or three times of stem water

sampling during each growth period. The average value of the partitioned E or T during one

growth period was used to represent the ET partitioning result in this period.

The final and initial soil water storage ($m_f$ and $m_i$) in Eq. (1) was calculated using the

measured depth-weighted volumetric soil water content. Meanwhile, the precipitation ($m_P$)

was obtained from meteorological observations, while the irrigation ($m_I$) was artificially





controlled and therefore measurable. The soil moisture near the bottom boundary remained
steady and generally below the field capacity throughout the experimental seasons. Therefore,
the amount of drainage ($m_D$) was neglected in this study. In addition, no runoff ($m_R$) was
observed during the field experiments.

The values of $\delta_i$ and $\delta_f$ are the depth-weighted $\delta^{18}$O averages for the whole soil profile

collected on and after the day of stem water sampling, respectively, while $\delta_P$ and $\delta_I$ are the
measured $\delta^{18}$O values of the precipitation and irrigation, respectively. The $\delta^{18}$O value of
evaporation ($\delta_E$) is estimated using the fraction factor $\alpha_{\text{liquid-vapor}} = (\delta_l +1000)/ (\delta_v +1000)$
(Wang et al., 2012). The evaporated water $\delta_v$ ($\delta_E$ in Eq. (3)) is assumed to be in isotopic
equilibrium with the soil water $\delta_l$ ($\delta_i$ in Eq. (3)). The value of $\alpha_{\text{liquid-vapor}}$ is given as 1.0102 at
an air temperature of 15 ℃ following Clark and Fritz (1997). As there is no fractionation in
the T processes of winter wheat (Wang and Yakir, 2000), the value of $\delta_T$ is determined using
the measured $\delta^{18}$O of stem water.
**2.4   MixSIAR model**
The MixSIAR Bayesian mixing model (v2.1.3) incorporating with dual stable water isotopes
($\delta$D and $\delta^{18}$O) was used to identify the water uptake sources of winter wheat. In field
experiments, precipitation or irrigation water infiltrated and finally mixed into the old soil
water. Groundwater could hardly contribute to crop water use (the average maximum rooting
depth was 2 m for winter wheat) under the condition of the deep water table depth (mean of
16 m below the soil surface). It can be supposed that soil water at different depths was
proportionally sourced by winter wheat. Four layers was divided as 0-20, 20-70, 70-150, and
150-200 cm depth along the 2-m soil profile in terms of their water isotopic compositions,



soil moisture contents and root distributions. The dual stable isotopes of the soil water in each
layer (raw source data) and of the stem water (mixture data) were input to the MixSIAR
model to quantify the main root water uptake depth. The Markov chain Monte Carlo (MCMC)
was used in the MixSIAR model for estimating the probability density functions of variables
as the MCMC was advantageous to estimate the entire distribution for each variable. The
MCMC parameter run length was set to "very long" to converge on the true posterior
distribution for each variable. The model error was evaluated using the SIAR (process and
residual). The estimated $5^{th}$, $25^{th}$, $50^{th}$, $75^{th}$, and $95^{th}$ percentiles of the posterior contributions
of each source described the distribution associated with the proportional contribution of each
source to winter wheat. The 50% percentile represented the median source contribution value
for each source.
**2.5   Data analysis**
The statistical analyses of the variation in each isotopic composition, soil moisture
distribution, ET component and associated fraction, and root water uptake pattern during each
season and treatment were all performed using a Statistical Package for the Social Sciences
(SPSS) 19.0 software package. The WUE was defined as the ratio between the crop yield and
the total ET from the greening to harvest season of winter wheat (Hussain et al., 1995). In this
study, the highest WUE value without an evident decrease in the grain yield was used as the
primary criterion with which to evaluate the optimal agricultural management practice.
Regression analyses of either $F_T$ or T with the LAI, crop yield and WUE were all performed
to investigate the relationships between the partitioning of ET and crop development.
**3   Results**





### 3.1   Isotopic compositions of different waters


The LMWL was established as $\delta D = 7.3\ \delta^{18}O + 3.6$ ($R^2 = 0.97$, $p < 0.01$) and $\delta D = 6.7\ \delta^{18}O +$
1.8 ($R^2 = 0.97$, $p < 0.01$) for the 2014 and 2015 experimental seasons, respectively (Fig. 1).
The smaller slope of the LMWL in 2015 than in 2014 was ascribed to a faster evaporation
rate of falling raindrops (Wang et al., 2010b). As shown in Fig. 1, the soil water isotopes
mainly fell below the LMWL, especially in 2015. The slope of the fitting line between $\delta D$
and $\delta^{18}O$ in soil water was lower in 2015 (2.8) than in 2014 (4.0). It indicated that the soil
water was more strongly evaporated in 2015.
According to two-way Analysis of variance (ANOVA), the isotopic profiles of the soil
water showed significant differences among the different layers and growth stages ($p < 0.05$).
The $\delta D$ and $\delta^{18}O$ values of the soil water in the surface layer (0-20 cm) were remarkably
enriched and indicated that the soil water isotopes had been subjected to extremely
evaporative fractionation. The soil water isotope values in the 0-20 cm layer were
significantly different from those in the other layers throughout the growing seasons ($p <$
0.05). The soil water isotopes in the 20-70 and 70-150 cm layers were intensively
fractionated since the jointing stage. No significant seasonal changes were detected in the
isotopic compositions of the soil water in the 150-200 cm layer, and they were similar to
those values of irrigation water (Fig. 1). The stem water isotopes were mainly concentrated
along the fitting line of the $\delta D$-$\delta^{18}O$ relationship in soil water (Fig. 1). The majority of the
stem water isotopes in 2014 matched well with the soil water isotopes in the 0-150 cm layer;
nevertheless, they were more enriched in 2015 and fell in the upper soil layer (0-70 cm).
Therefore, the maximum root water uptake depth of winter wheat probably approached 150





cm during the experimental seasons.

### 3.2   Seasonal changes in soil water storage and ET

Approximately 127.9 mm of the soil water storage in the profile of 0-200 cm was consumed
on average throughout the whole season from wheat greening to harvest. Approximately 92%
of this consumption occurred in the 0-150 cm layer (Fig. 2). The slight change in the soil
moisture within the 150-200 cm layer was consistent with the small variation in the soil water
isotopic compositions in the same layer (Figs. 1-2). A greater amount of soil water storage
(with a mean value of 35.2 mm) was consumed in 2015 than in 2014, primarily within the
0-70 cm layer (Fig. 2). The largest reduction in the soil water storage during the 2015 season
occurred during the jointing-heading period (98.1 mm), and this reduction accounted for 67.4%
of the total loss. Among the five treatments, T4 showed the highest consumption of soil water
storage during the 2014 (151.8 mm) and 2015 (174.5 mm) seasons. Sufficient irrigation
during treatment T5 in 2014 led to the smallest observed reduction in the soil water storage
(80.3 mm). However, the reduction in the soil water storage under treatment T5 notably
increased to 143.4 mm in 2015. This was primarily caused by severe reductions in the soil
water storage during the jointing-heading and filling-harvest periods without irrigation under
dry climatic conditions.

The total ET throughout the season from wheat greening to harvest was a mean of 292.8

mm with a standard deviation (SD) of 38.2 mm (Table 4). The total ET increased on average
by 45.3 mm in 2015 relative to 2014, and this was in general agreement with the observed
increment of soil water consumption in 2015. The reference agricultural management practice
(T5) remarkably raised the crop water consumption in terms of the largest ET value in the



growing seasons of both 2014 (304.0 mm) and 2015 (377.3 mm). The daily mean ET was
significantly different ($p < 0.01$) among the four growth periods with values of 3.0, 5.0, 5.4,
and 4.0 mm d$^{-1}$ in the greening-jointing, jointing-heading, heading-filling, and filling-harvest
periods, respectively. The higher daily mean ET flux during the mid-season stage (i.e., the
jointing-filling stage) was mainly due to a higher LAI and an increased biomass.
**3.3   Seasonal variations in ET partitioning**
The seasonal variations in the partitioning of ET are shown in Fig. 3. The daily mean T
changed significantly among the different periods during the experimental seasons of both
2014 and 2015 ($p < 0.01$) (Fig. 3). The daily mean T was evidently small (2.0 mm d$^{-1}$) during
the early growth stage of greening-jointing and reached a high level during the
jointing-heading and heading-filling periods (4.4 and 4.6 mm d$^{-1}$, respectively), after which it
declined moderately to 3.4 mm d$^{-1}$ until the winter wheat harvest. In contrast to T, a
substantial seasonal variance in the daily mean E was detected only in 2014 with values of
1.1, 0.3, 0.8, and 0.6 mm d$^{-1}$ during the greening-jointing, jointing-heading, heading-filling,
and filling-harvest periods, respectively (Fig. 3). In 2015, the differences in the daily mean E
among the four periods were small with an average value of 0.8 mm d$^{-1}$. A significant
difference in the daily mean E between 2014 and 2015 occurred in the jointing-heading
period, as it increased to 1.0 mm d$^{-1}$ in 2015 due to severe drought induced by little
precipitation and the lack of irrigation.
The values of $F_T$ varied widely from 0.51 to 0.98 during the individual growth periods
under different treatments. The mean values of $F_T$ were 0.65, 0.88, 0.84, and 0.85 during the
greening-jointing, jointing-heading, heading-filling, and filling-harvest periods, respectively





(Fig. 4). These results demonstrate that the average $F_T$ during the jointing-heading period of
the 2015 season (0.82) was much lower than that of the 2014 season (0.94). In particular, the
decreasing of $F_T$ in the jointing-heading period from 2014 to 2015 was evident under
treatments T1, T2, and T5 with reductions of 0.25, 0.14 and 0.16, respectively. Moreover, the
performance of $F_T$ in each growth period was notably distinct among the different treatments
(Fig. 4). Compared with the mean level of $F_T$ for all of the treatments, $F_T$ was 16.9% larger
during the greening-jointing period but significantly less (17.6%, $p < 0.05$) during the
filling-harvest period under treatment T1. The T5 under the reference agricultural
management practices had the smallest $F_T$ during greening-jointing period in 2015.
The $F_T$ value during the whole season had an average value of 0.82, and it did not vary
significantly among the seasons and treatments (with an SD of 0.03, $p > 0.05$) (Table 4). The
T during the jointing-heading and heading-filling periods ($T_{jf}$) accounted for approximately
50% of the seasonal ET, and $T_{jf}$ exhibited a significant positive linear correlation with the
total ET ($R^2 = 0.82$, $p < 0.01$). Therefore, $T_{jf}$ played a critical role in determining the
variations in ET throughout the growing season. Fig. 4 demonstrates that the value of $T_{jf}$ was
greatly different between the two experimental seasons. The average $T_{jf}$ was 34.9 mm more
in 2015 than in 2014, occupying 76.9% of the increment (45.3 mm) in the total ET from 2014
to 2015. Furthermore, both the largest $T_{jf}$ and the highest total ET were observed under the
reference treatment (T5) in 2015.
Fig. 5 reveals that $F_T$ increased with the LAI and varied around an asymptotic value of
0.87 when the LAI was between 2.7 and 8.7. The value of $F_T$ was low with an average of
0.64 during the early growth stage (i.e., the greening-jointing period) with small LAI values





(0.7-2.0), while T was the predominant partition in the ET during the mid-growing season
when the LAI exceeded 2.7 (Fig. 5). However, $F_T$ reached a maximum of 0.78 with a small
LAI (1.11) under treatment T1 during the 2015 season. The seasonal changes in $F_T$ can be
effectively described as a power-law function of the LAI ($F_T = 0.61$ LAI $^{0.21}$, $R^2 = 0.66$,
$p<0.01$) for winter wheat (Fig. 5). This implied that crop development played a major role in
driving the contribution of T to ET and that the LAI could provide insights into estimating the
variability in $F_T$ throughout the growing season of winter wheat.
**3.4    Seasonal variations in root water uptake patterns**
The contributions from soil water in different layers to root water uptake estimated using the
MixSIAR model are shown in Fig. 6. The average contributions of soil water to winter wheat
within the 0-20, 20-70, 70-150, and 150-200 cm layers during the 2014 growing season were
28.7%, 30.0%, 26.9%, and 14.4%, respectively. The root water uptake depth tended to
become deeper with crop development (Fig. 6a). Winter wheat mainly acquired soil water
from the 0-20 cm (63.6%), 20-70 cm (67.9%), 70-150 cm (54.4%), and 70-150 cm (39.8%)
layers during the greening-jointing, jointing-heading, heading-filling, and filling-harvest
growth periods, respectively. The 150-200 cm layer contributed a certain amount of soil
water to winter wheat since the jointing-heading period and reached a maximum mean
proportion of 27.2% in the filling-harvest period.

As shown in Fig. 6, higher quantities of shallow soil water were taken up by winter wheat

in 2015, particularly within the top layer (0-20 cm) with average contributions of 28.7% and
42.6% in 2014 and 2015, respectively. The predominant water uptake depth was 0-20 cm in
both the greening-jointing period (70.4%) and the heading-filling period (63.4%) in 2015.



During the jointing-heading period, the limited water supply and high T rates remarkably
promoted the average contribution of deep soil water with values of 32.2% and 23.5% in the
70-150 cm and 150-200 cm layers, respectively. Meanwhile, winter wheat took up
significantly more soil water from the 20-70 cm layer (54.9%) during the filling-harvest
period than during the other periods ($p < 0.01$).
**3.5   Relationships between grain yield and WUE with seasonal partition of ET**
The crop yield and WUE of winter wheat throughout the growing season for each treatment
are shown in Table 4. The mean grain yield was 6759.9 kg ha$^{-1}$ with an SD of 478.5 kg ha$^{-1}$.
Compared with the reference treatment, the T1, T2, and T3 treatments reduced the grain yield
by more than 10%, while treatment T4 raised the yield by 0.9% for the 2014 season.
Treatment T5 exhibited the lowest grain yield in the 2015 season, while the other treatments
showed a 15.6% increment on average. The mean WUE was 24.9 kg ha$^{-1}$ mm$^{-1}$ in the 2014
season and 21.9 kg ha$^{-1}$ mm$^{-1}$ in the 2015 season. The variability in the WUE among the
different treatments was greater in 2015 than in 2014. The maximum WUE was observed
under T1, whereas T5 showed the smallest WUE in each season.
The results demonstrate that both the grain yield and the WUE were not significantly
correlated with the $F_T$ throughout the experimental season ($p > 0.05$). The observed grain
yield positively increased with $T_{jf}$ in 2014, whereas it decreased with $T_{jf}$ in 2015. The grain
yield reduced remarkably with excessive values of $T_{jf}$ (205.8 mm) under treatment T5 in
2015 (Fig. 7 and Table 4). A significant quadratic relationship was found between the grain
yield and $T_{jf}$ ($R^2 = 0.77$, $p < 0.01$) (Fig. 7). The peak grain yield was 7062.6 kg ha$^{-1}$ at a $T_{jf}$
value of 155.8 mm along this fitting curve. The WUE also had a significant quadratic





418 correlation with $T_{jf}$ ($R^2 = 0.87$, $p < 0.01$) (Fig. 7). The peak WUE along the fitting curve was

419 24.9 kg ha$^{-1}$ mm$^{-1}$ with a $T_{jf}$ value of 117.5 mm. Most $T_{jf}$ values were larger than this critical

420 value except for that under treatment T3 in 2014. As $T_{jf}$ exceeded 117.5 mm, the WUE

421 declined to a minimum value of 16.0 kg ha$^{-1}$ mm$^{-1}$ with a continuous increase in $T_{jf}$ (Fig. 7).

422 This suggested that the magnitude of $T_{jf}$ controlled both the grain yield and the WUE for

423 winter wheat in this region.

424 **4 Discussion**

425 **4.1 Influencing factors of seasonal variations in ET partitioning**

426 The daily T flux estimated in this study (ranging from 2.0 to 4.6 mm d$^{-1}$) was similar to those

427 of 1.02-4.91 mm d$^{-1}$ (Zhang et al., 2011) and 0.8-4.5 mm d$^{-1}$ (Liu et al. 2002) under surface

428 irrigation in the NCP determined via the isotope/eddy covariance and

429 weighing/micro-lysimeters methods, respectively. E is a significant component of ET,

430 especially when the LAI is low. The seasonal $F_E$ (0.18) was also in accordance with the value

431 of 0.23 reported by Liu et al. (2002). The $F_E$ value calculated in our study reached up to 0.35

432 during the greening-jointing period, which is consistent with the estimation of 0.30 from

433 Zhang et al. (2011). This study indicated that the seasonal changes in $F_T$ could be effectively

434 described through a power-law function of the LAI. This relationship was similar to those

435 obtained in recent studies at both the global and the field scale (Wang et al., 2014; Wei et al.,

436 2015; Wu et al., 2017; Lu et al., 2017). The strong correlation between $F_T$ and LAI confirmed

437 that $F_T$ was controlled by LAI at seasonal timescale (Wang and Yamanaka, 2014; Wang et al.,

438 2014; Wei et al., 2015; Wei et al., 2018). When LAI was less than 2.7, $F_T$ increaseed

439 significantly with crop development in the early growing season and then it converged



towards a stable value beyond LAI of 2.7. This threshold of LAI (2.7) to distinguish the two
different changing trends of $F_T$ agreed well with the values of 2.5 and 3.0 reported in Wei et
al. (2018) and Kang et al. (2003), respectively. $F_T$ has been shown to reach a high level (0.90
for agricultural systems at the global scale and 0.58 for a paddy field), even under low LAI
conditions (Wang et al., 2014; Wei et al., 2015). In this study, the estimated $F_T$ reached up to
0.78 with a small LAI (1.11) under treatment T1 during the 2015 growing season. The above
comparisons indicate that the ET partitioning results in this study are reliable.
Besides LAI, $F_T$ was influenced greatly by soil moisture, especially the topsoil moisture
in 0-20 cm depth. Previous studies indicated that $F_T$ generally decreased with increasing
topsoil moisture due to increase of E under the same LAI conditions (Liu et al., 2002; Yu et
al., 2009; Wei et al., 2018). A negative linear correlation was found between $F_T$ and surface
soil water content ($\theta_v$) when LAI was about 1.8 ($F_T$=-1.38$\theta_v$+1.0, $R^2$=0.98, $p$<0.01) during
greening-jointing period in our experiments. It was suggested that keeping surface soil dry
without affecting the crop ET was an important way to reduce E in the early growing season
(Liu et al., 2002). However, increasing $\theta_v$ remarkably increased $F_T$ at LAI of about 4.0
($F_T$=1.83$\theta_v$+0.6, $R^2$=0.74, $p$<0.01) during filling-harvest period.
Factors controlling E and T were coupled in ways to affect $F_T$ under dry climate condition
particularly during jointing-heading period in 2015. Adequate rainfall falling during
greening-jointing period (35 mm) led to larger $\theta_v$ at the early stage in jointing-heading period
(mean of 0.19). Great availability of soil moisture in the topsoil increased water contribution
to E. Furthermore, the strong atmosphere demand remarkably promoted E at late stage of the
jointing-heading period (Zhao et al., 2018). This resulted in the significant increase of the E



rate to 1.0 mm d$^{-1}$ in the period. It was the topsoil moisture greatly influencing E, while the
water used for T came from the whole root zone. Although continuous E caused the extreme
consumption of surface soil moisture in the drought period, soil water storage in the
subsurface layers could meet T requirement of crop. Soil water in the deep layers could move
into the upper dry layer via hydraulic lift through the process of root water uptake (Jha et al,
2017; Li et al., 2010). High water uptake from deep layers (32.2% and 23.5% in the 70-150
cm and 150-200 cm layers, respectively) may improve the plant leaf water content and
maintain T rates and dry matter production.

Distributions of soil moisture and root water uptake patterns were significantly

influenced by different irrigation and fertilization treatments especially in dry seasons. More
frequently irrigated treatments have previously been reported to have more roots in the
surface layer than less irrigated treatments (Zhang et al., 2004). Meanwhile, nitrogen
fertilizers stimulated root growth near the soil surface and abundant soil nitrogen content
might increase the drought resistance of the root system under water limited condition
(Kmoch et al., 1957; Carvalho and Foulkes, 2013). Ma and Song (2016) showed that the soil
water contribution had a significantly positive and linear relationship with the proportion of
root length. Therefore, plant primarily took up soil water from the top layer (0-20 cm) under
the T4 and T5 treatments even though the climate was dry during jointing-heading period in
2015. However, nitrogen deficiency promoted root growth in the deep soil layer (150-200 cm)
and increased water adsorption by 43.1% under the T1 and T3 treatments. Previous studies
demonstrated that plants growing in drier environments with soil water deficit in the surface
layer have deeper root systems and more branched seminal roots (Morita et al., 1997; Zhang



et al., 2004; Jha et al., 2017). This confirmed that over 80% of plant water took up soil water
from the 70-200 cm layer under the less irrigation treatments of T1 and T2. When soil water
near the surface was replenished by irrigation, the extraction depth returned to the surface
layer (such as in heading-filling period) and subsequently moved downward again until
harvest of winter wheat.
**4.2    Application for optimizing water management practices**
With the abovementioned ET partitioning results and the fitted WUE-$T_{jf}$ and Yield-$T_{jf}$ curves,
the irrigation and fertilization schedules were optimized. As shown in Fig. 7, the value of $T_{jf}$
should be controlled between 117.5 and 155.8 mm to obtain both a high grain yield and a
high WUE. The $T_{jf}$ under treatments T1, T2, T4, and T5 in 2014 and that under treatment T1
in 2015 acquired in this study were within this range. An additional irrigation of 140 mm was
required for treatment T5 compared with the T1. Although the T1 treatment in 2014 had a
larger WUE, its grain yield was diminished by 8.5% relative to 2015. Therefore, the T1
treatment in 2015 optimally improved the WUE and maintained a high grain yield. The
optimal irrigation and fertilization schedules can be determined as two irrigations during the
greening-jointing (20 mm) and heading-filling (80 mm) periods and one fertilization (105 kg
ha$^{-1}$ N) during the greening-jointing period. The designed wetting layer should be controlled
at depths of 0-70 cm because wheat primarily sourced soil water from the 0-70 cm layer
during the experimental seasons. This practice could make better use of the deep soil water
storage and avoid deep percolation compared with a traditional wetting depth of 100 cm
(Zhang et al., 2011).

The obtained optimal agricultural management practice is supported by previous studies



in the NCP. The first small irrigation was applied mainly to the top soil to ensure that the
fertilization was distributed evenly throughout the plot. The irrigation in the early growth
stage reduced the grain yield because it enhanced the development of non-functional tillers,
which consume the reserved nutrients (Sun et al., 2006). Wang et al. (2014) and Lu et al.
(2017) reported that water loss via E could be much higher during the vegetative stage than
during later growth stages. Reducing the irrigation amount during the greening-jointing
period could increase the depletion of deep soil water, and it was definitely necessary to
improve the WUE of winter wheat. Meanwhile, the heading growth period was extremely
sensitive to water stresses, and irrigation is strongly recommended during this period (Zhang
et al., 2003; Li et al., 2005; Shang and Mao, 2006). Therefore, the obtained optimal schedule
in this study is appropriate, as it could conserve 140 mm of irrigation water and 105 kg ha$^{-1}$ N
of fertilizer with respect to the reference practices.

**4.3    Further scopes of this study**

The ET of winter wheat was partitioned effectively into E and T using the isotope mass
balance and water balance methods. The partitioning of ET changed between different
irrigation and fertilization schedules and various crop development stages. The evaluation
using isotopic data presented a quantitative correlation between seasonal change in the $F_T$ and
the crop development of LAI. The relationships among the grain yield and WUE with the $T_{jf}$
were discovered. This isotope-based method provided insights into clarifying the
hydrological processes in field ecosystem and optimizing water and nitrogen management
practices. Nevertheless, several issues still need further investigation. First, although the
interception flux is often neglected in many partitioning works, it indeed is a component of





ET besides E and T and need further estimation. Second, the water flux at the bottom
boundary of the soil profile was generally neglected in the estimation of ET due to small
changes in soil moisture during winter wheat growing season under limited irrigation in the
NCP (Zhang et al., 2003; Li et al., 2005; Li et al., 2010). However, drainage should be
accurately evaluated by the Darcy's law when soil moisture at the bottom boundary is above
field capacity. Third, as calculated from the MixSIAR model, each soil layer had a different
contribution to the root water uptake. Incorporating these contributions into the isotopic mass
balance equation can reflect the variation in the gradient of the isotopic profile. Finally,
high-frequency measurements of isotopic composition of soil, stem and gas water will
improve understanding the seasonal variation in ET partitioning.
**4    Conclusions**
In this study, the isotope mass balance were coupled with water balance methods for the
partitioning of evapotranspiration (ET) into crop transpiration (T) and soil evaporation (E) of
winter wheat under different irrigation and fertilization treatment schemes during 2014-2015
in Beijing, China. The fraction of T in ET ($F_T$) showed averages of 65.4%, 87.7%, 83.8%,
and 84.9% in the greening-jointing, jointing-heading, heading-filling, and filling-harvest
periods, respectively. The performance of $F_T$ was notably distinct among the different
treatments in each growing period. However, the value of $F_T$ throughout the season from
greening to harvest did not vary significantly among the seasons and treatments ($p > 0.05$)
and had an average value of 0.82. The seasonal change in $F_T$ could be effectively described as
a power-law function of the LAI ($F_T = 0.61$ LAI $^{0.21}$, $R^2 = 0.66$, $p<0.01$). Winter wheat mainly
utilized soil water from the 0-20 cm (67.0%), 20-70 cm (42.0%), 0-20 cm (38.7%), and 20-70



cm (34.9%) layers during the greening-jointing, jointing-heading, heading-filling, and
filling-harvest periods, respectively. The main root water uptake depth increased with the
crop development in 2014, whereas it was mostly concentrated within the 0-70 cm layer in
2015. $F_T$ was not significantly correlated with the grain yield and WUE ($p > 0.05$), and the
total T during the jointing-heading and heading-filling periods ($T_{jf}$) had a significant
quadratic relationship with the grain yield and WUE ($p < 0.01$). In order to obtain the optimal
crop yield, 20 mm and 80 mm of irrigation water during the greening-jointing and
heading-filling periods, and 105 kg ha$^{-1}$ N of fertilization during the greening-jointing period
were needed. The designed wetting layer should be controlled at depths of 0-70 cm. This
study demonstrated the roles of seasonal ET partitioning obtained via isotope-based methods
in determining the crop development and improving the WUE, and the findings acquired
herein have important implications on irrigation and fertilization management.

*Acknowledgements.* This work was supported by the National Natural Science Foundation of
China (Grant No. 41671027; 41730749). We thank Ningxia Sun for field data collection and
Dr. Baozhong Zhang and Lihu Yang for their assistance in field and laboratory experiments.

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



**Figure captions**
**Fig. 1.** $\delta$D-$\delta^{18}$O relationship for the water samples and the local meteoric water line (LMWL)

in the (a) 2014 and (b) 2015 growing seasons.

**Fig. 2.** Seasonal variations in the soil moisture under the (a) T1, (b) T2, (c) T3, (d) T4, and (e)

T5 treatments.

**Fig. 3.** Mean daily evaporation (E) and transpiration (T) rates of winter wheat during each

growth period in the 2014 and 2015 seasons (mean $\pm$SD). ○ and * represent outliers

with a 1.5$\times$interquartile range (IQR) and a 3IQR, respectively.

**Fig. 4.** Seasonal variations in T, E, and the fraction of transpiration within evapotranspiration

($F_T$) in winter wheat for each treatment during (a) 2014 and (b) 2015.

**Fig. 5.** Relationship between the fraction of transpiration within evapotranspiration ($F_T$) and

the leaf area index (LAI).

**Fig. 6.** Proportions of the soil water contribution to winter wheat during each growth stage in

(a) 2014 and (b) 2015 (mean $\pm$SD). ○ and * represent outliers with a 1.5$\times$

interquartile range (IQR) and a 3IQR, respectively.

**Fig. 7.** Relationships among the grain yield and water use efficiency (WUE) with the total

transpiration during the jointing-heading and heading-filling periods ($T_{jf}$).





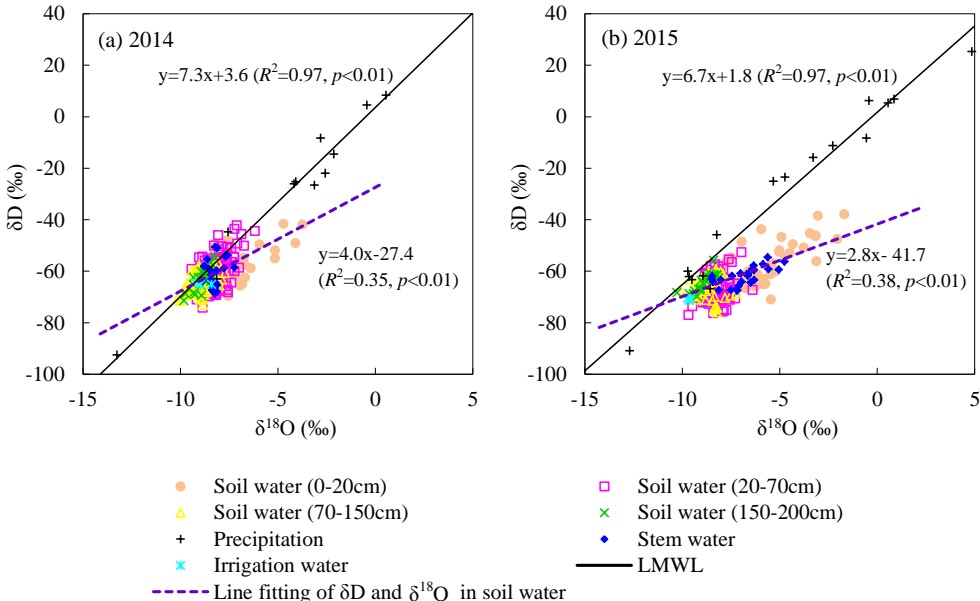

**Fig.1**. δD-δ$^{18}$O relationship for the water samples and the local meteoric water line (LMWL) in the (a)

2014 and (b) 2015 growing seasons.




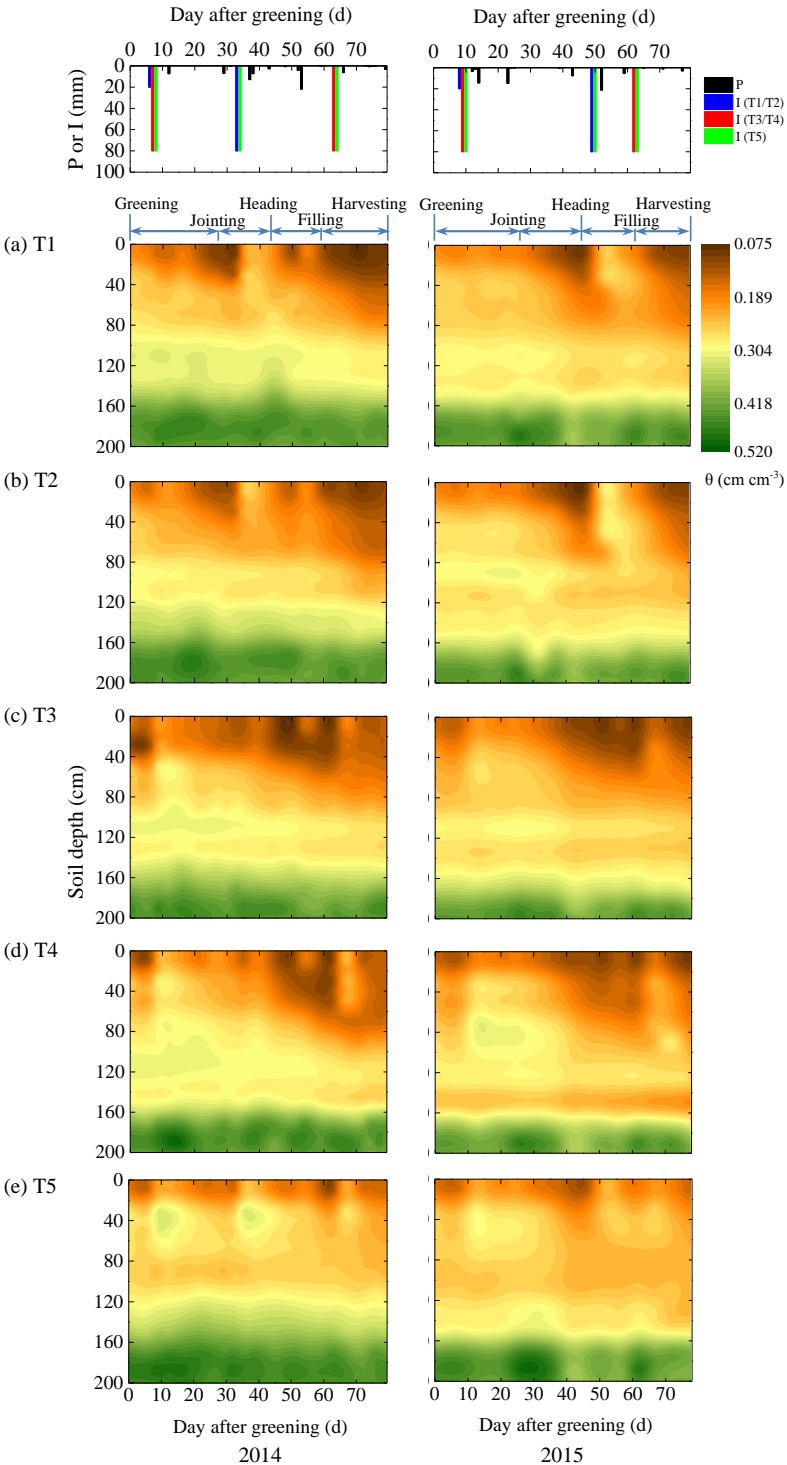

**Fig.2.** Seasonal variations in the soil moisture under the (a) T1, (b) T2, (c) T3, (d) T4, and (e) T5

treatments.




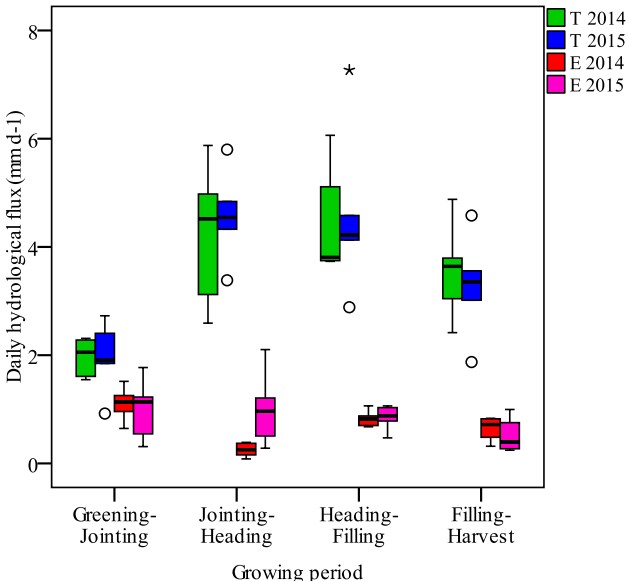

**Fig.3.** Mean daily evaporation (E) and transpiration (T) rates of winter wheat during each growth

period in the 2014 and 2015 seasons (mean ±SD). ○ and * represent outliers with a 1.5×

interquartile range (IQR) and a 3IQR, respectively.





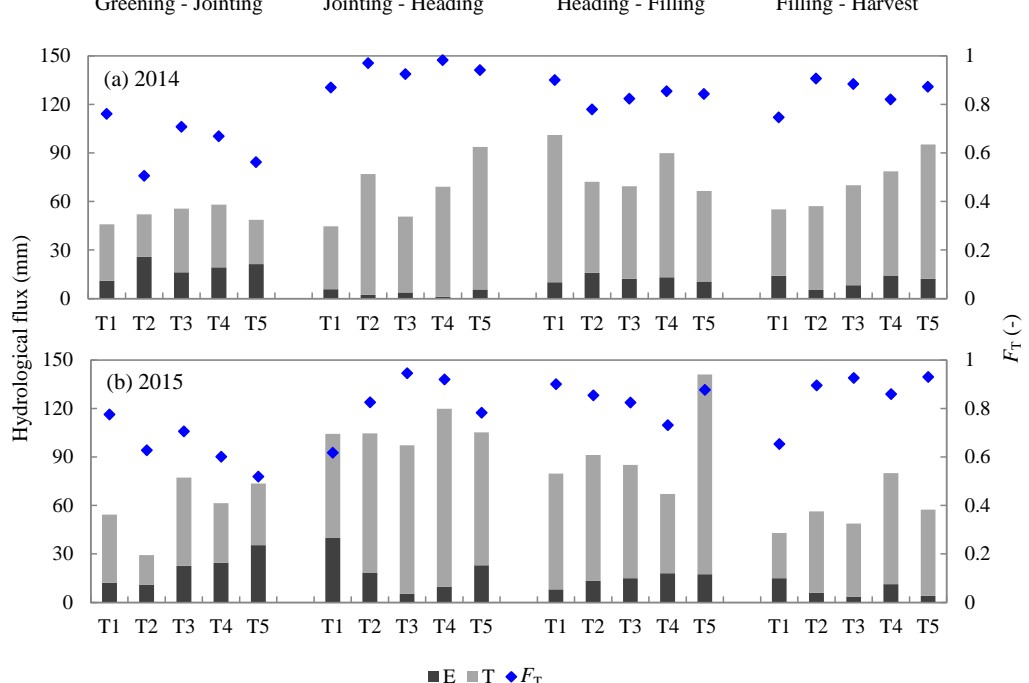

**Fig.4.** Seasonal variations in T, E, and the fraction of transpiration within evapotranspiration ($F_T$) in winter wheat for each treatment during (a) 2014 and (b) 2015.





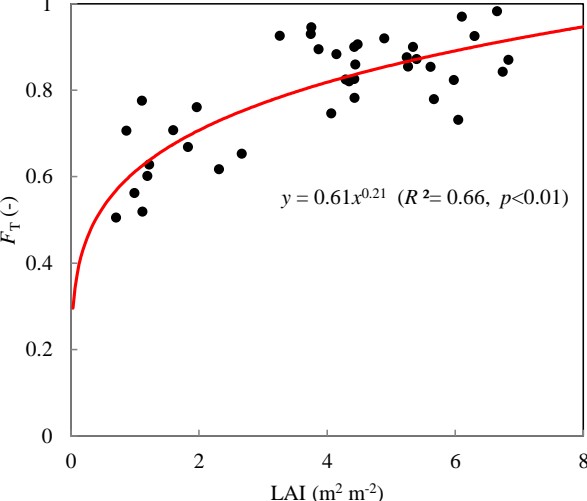

$y = 0.61x^{0.21}$  ($R^2 = 0.66$, $p < 0.01$)

**Fig.5.** Relationship between the fraction of transpiration within evapotranspiration ($F_T$) and the leaf area index (LAI).





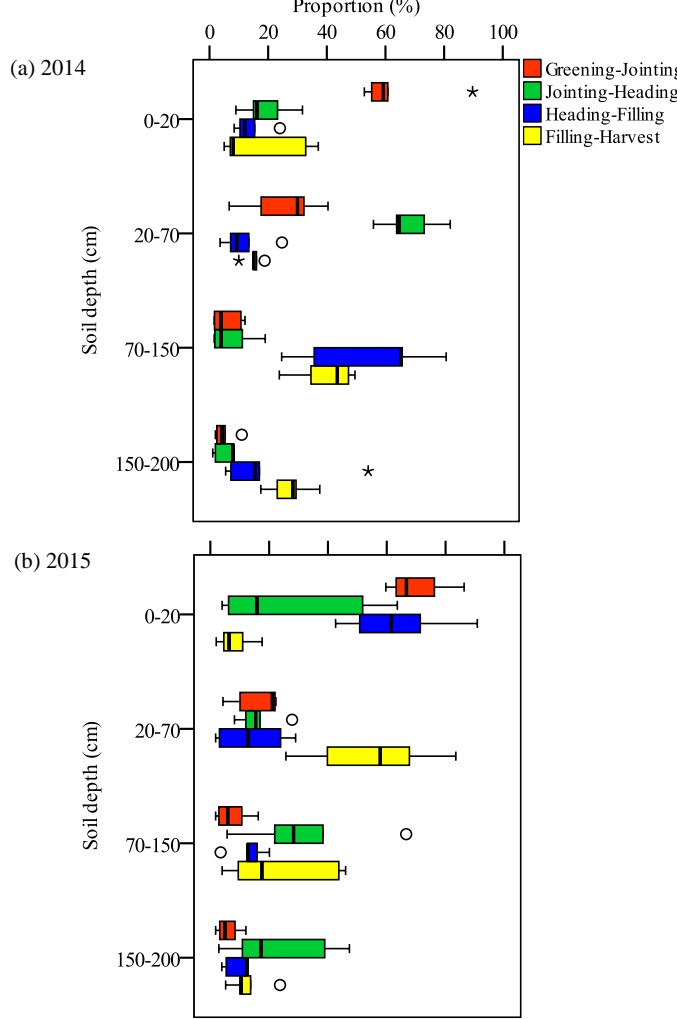

**Fig.6.** Proportions of the soil water contribution to winter wheat during each growth stage in (a) 2014 and (b) 2015 (mean ±SD). ○ and * represent outliers with a 1.5× interquartile range (IQR) and a 3IQR, respectively.



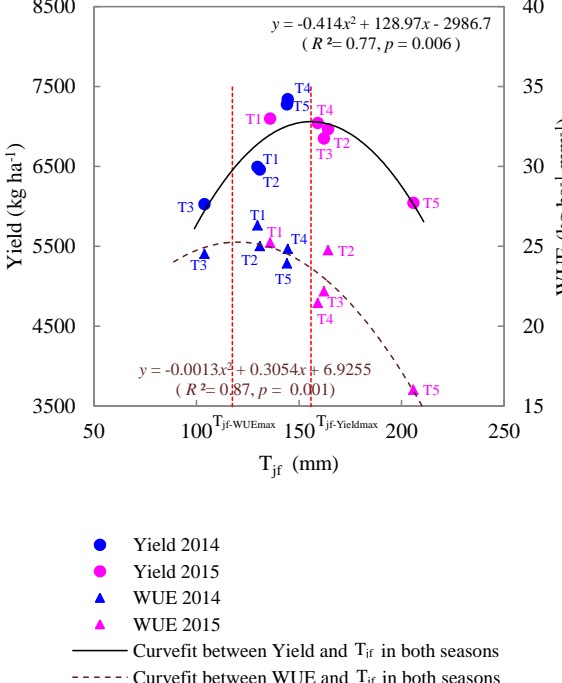

**Fig.7.** Relationships among the grain yield and water use efficiency (WUE) with the total transpiration during the jointing-heading and heading-filling periods ($T_{jf}$). The two vertical red dashed lines represented the $T_{jf}$ under the maximum WUE ($T_{jf\text{-}WUEmax}$) and Yield ($T_{jf\text{-}Yieldmax}$) conditions.





**Table 1.** Physical and chemical properties of the soil profile at the experimental site.

| Depth (cm) | Particle size (%) | | | Soil texture | Bulk density (g cm$^{-3}$) | $\theta_s$ (cm$^3$ cm$^{-3}$) | $K_s$ (cm d$^{-1}$) | OC ( g kg$^{-1}$) | EC ($\mu$S cm$^{-1}$) | pH | NH$_4^+$-N (mg kg$^{-1}$) | NO$_3^-$-N (mg kg$^{-1}$) |
|---|---|---|---|---|---|---|---|---|---|---|---|---|
| | Sand | Silt | Clay | | | | | | | | | |
| 0-20 | 58.8 | 33.2 | 8.0 | Sandy loam | 1.56 | 0.41 | 8.41 | 8.03 | 111.70 | 8.15 | 7.0 | 98.9 |
| 20-120 | 65.3 | 26.7 | 8.0 | Sandy loam | 1.48 | 0.42 | 10.04 | 3.87 | 109.12 | 8.61 | 5.9 | 17.9 |
| 120-180 | 68.2 | 29.2 | 2.7 | Sandy loam | 1.45 | 0.45 | 7.45 | 1.52 | 87.60 | 8.66 | 6.3 | 20.2 |
| 180-200 | 32.0 | 51.0 | 17.0 | Silt loam | 1.25 | 0.51 | 0.66 | 5.41 | 161.80 | 8.27 | 4.4 | 19.0 |

Note: OC: Organic C; $\theta_s$: Saturated water content, $K_s$ : Saturated hydraulic conductivity, EC: Electric conductivity, pH: Potential of hydrogen, NH$_4^+$-N: Ammonia nitrogen, NO$_3^-$-N: Nitrate nitrogen



**Table 2.** Irrigation (I) and fertilization (N, as urea) schedules for each treatment of the winter wheat during the experimental growing seasons. (units: mm for I and kg ha$^{-1}$ for N)

| Season | Treatment | Greening-Jointing | | Jointing-Heading | | Heading-Filling | | Filling-Harvest | | Total | |
|--------|-----------|---|---|---|---|---|---|---|---|---|---|
| | | I | N | I | N | I | N | I | N | I | N |
| 2014 | T1 | 20 | 105 | 80 | – | – | – | – | – | 100 | 105 |
| | T2 | 20 | 315 | 80 | – | – | – | – | – | 100 | 315 |
| | T3 | 80 | 105 | – | – | – | – | 80 | – | 160 | 105 |
| | T4 | 80 | 315 | – | – | – | – | 80 | – | 160 | 315 |
| | T5 | 80 | 210 | 80 | – | – | – | 80 | – | 240 | 210 |
| 2015 | T1 | 20 | 105 | – | – | 80 | – | – | – | 100 | 105 |
| | T2 | 20 | 315 | – | – | 80 | – | – | – | 100 | 315 |
| | T3 | 80 | 105 | – | – | – | – | 80 | – | 160 | 105 |
| | T4 | 80 | 315 | – | – | – | – | 80 | – | 160 | 315 |
| | T5 | 80 | 210 | – | – | 80 | – | 80 | – | 240 | 210 |

Note: The T5 treatment represents conventional practice, and "−" shows no irrigation or fertilization applied.





**Table 3.** Evapotranspiration (ET) partitioning, grain yield, and water use efficiency (WUE) of the winter wheat under each treatment.

| Season | Treatment | ET (mm) | T (mm) | E (mm) | $T_{jf}$ (mm) | $F_T$ (-) | Yield (kg ha$^{-1}$) | WUE (kg ha$^{-1}$ mm$^{-1}$) |
|---|---|---|---|---|---|---|---|---|
| 2014 | T1 | 246.8 | 205.8 | 41.0 | 129.8 | 0.83 | 6493.2 | 26.3 |
| | T2 | 258.4 | 208.9 | 49.5 | 130.9 | 0.81 | 6461.8 | 25.0 |
| | T3 | 245.7 | 205.1 | 40.6 | 103.9 | 0.83 | 6026.0 | 24.5 |
| | T4 | 295.6 | 247.7 | 47.9 | 144.5 | 0.84 | 7341.6 | 24.8 |
| | T5 | 304.0 | 254.4 | 49.6 | 144.1 | 0.84 | 7276.8 | 23.9 |
| | Mean | 270.1 | 224.4 | 45.7 | 130.6 | 0.83 | 6719.9 | 24.9 |
| | SD | 27.7 | 24.5 | 4.5 | 16.5 | 0.01 | 569.2 | 0.9 |
| 2015 | T1 | 281.3 | 206.2 | 75.1 | 136.0 | 0.73 | 7096.5 | 25.2 |
| | T2 | 281.5 | 233.0 | 48.5 | 164.2 | 0.83 | 6965.5 | 24.7 |
| | T3 | 308.7 | 261.9 | 46.8 | 162.1 | 0.85 | 6848.9 | 22.2 |
| | T4 | 328.4 | 264.9 | 63.5 | 159.2 | 0.81 | 7044.5 | 21.5 |
| | T5 | 377.3 | 297.3 | 80.0 | 205.8 | 0.79 | 6044.5 | 16.0 |
| | Mean | 315.4 | 252.7 | 62.8 | 165.5 | 0.80 | 6800.0 | 21.9 |
| | SD | 39.9 | 34.5 | 15.1 | 25.2 | 0.04 | 432.5 | 3.7 |

Note: $T_{jf}$ means the sum of transpiration in jointing-heading and heading-filling periods, and $F_T$ indicates the fraction of crop transpiration in evapotranspiration (T/ET).