# Peer review of "Seasonal variability in evapotranspiration partitioning and its"

_Hydrology and Earth System Sciences, 2018_

## Referee Comment (RC1) · Anonymous Referee #1 · 13 Jul 2018

Review for "Seasonal variability in evapotranspiration partitioning and its relationship with crop development and water use efficiency of winter wheat"

**Abstract**
Lines 21-22: Specify the definition of water use efficiency used in this study.

Lines 24-25: Not clear. What do you mean by ranging from 0.51 to 0.98? Seasonal variations of $F_T$ for all different treatments?

Lines 31-32: How can you increase/decrease $F_E$ during different periods? In addition, there should be an optimal range for $F_E$ during different periods. I do not think increasing $F_E$ will definitely result in higher WUE.

**Introduction**
Lines 54-55: You should also introduce several recently proposed ET partitioning methods, such as
Zhou, Sha, et al. "Partitioning evapotranspiration based on the concept of underlying water use efficiency." *Water Resources Research* 52.2 (2016): 1160-1175.

Scott, Russell L., and Joel A. Biederman. "Partitioning evapotranspiration using long-term carbon dioxide and water vapor fluxes." *Geophysical Research Letters* 44.13 (2017): 6833-6840.

Line 74: I do not agree with "precisely". The isotope method is better suited to more arid environments with sparse vegetation, and large uncertainty may exist in agricultural systems, see
Griffis, Timothy J. "Tracing the flow of carbon dioxide and water vapor between the biosphere and atmosphere: A review of optical isotope techniques and their application." *Agricultural and Forest Meteorology* 174 (2013): 85-109.

Lines 100-101: What kinds of possibilities? Could you point out the possibilities more clearly and provide relevant references?

Lines 110-112: add reference

Lines 142-143: Have you applied the results to practices? Or do you mean the results can be applied to practices?

**Methods**
Line 220: Only one ET partitioning method was used.

Section 2.3: Did you apply the partitioning method based on isotopic composition of d18O only? I find you observed both dD and d18O in different water pools.

**Results**
Line 287: What is LMWL? I find this is the first appearance of the acronym, so you should add detailed explanation or definition.

Fig. 1. It is not clear in which experiment the data shown in Fig. 1 were observed. Do the results vary in different experiments?

Line 326 and other places: No Table 4 in this study. I suppose it is Table 3. Is 292.8mm the mean value across all treatments in both 2014 and 2015?

Lines 363-366: The linear correlation does not measure how strong $T_{jf}$ contributes to variations in ET. Please use variance analysis instead.

Fig. 5: In which experiment do you get the $F_T$ and LAI data? Does the relationship vary in different experiments?

**Discussion**
Line 435: scales

Lines 442-444: Could you explain more why high $F_T$ exists even when LAI is low? Are there any mechanisms or some uncertianties in the observed data? Many studies show that the isotope method would obtain higher values of $F_T$ than other ET partitioning methods.

Lines 496-500: I do not agree with "optimally" used here. The results just compared several different treatments in 2014 and 2015. Although grain yield and WUE are higher in the T1 treatment in 2015, it is not optimal since there would be better treatments that were not considered in this study. Since different treatments were used in 2014 and 2015, it is not clear to what extent the increase of grain yield and WUE is attributed to different treatment, since different environmental conditions also play a part in different years.

Lines 500-503: How can you control the wetting layer? It seems that the wetting layer is not controlled to 0-70cm in this study. So the irrigation scheme can be further improved? This also contradicts to the so-called optimal agricultural management practice.

Section 4.3: Please add more limitations of this study, especially ET estimation and retrieval of isotopes in agricultural ecosystems, see Griffis (2013). In addition, the section title "further scopes of this study" is not appropriate, use "limitations of this study".

---

## Referee Comment (RC2) · Anonymous Referee #2 · 12 Aug 2018

Review of " Seasonal variability in evapotranspiration partitioning and its relationship with crop development and water use efficiency of winter wheat", submitted to HESS by Ma et al. Paper number: hess-2018-234

Summary: This manuscript used an isotope tracing technique based on water/mass balance to partition ET and quantify the root water uptake sources of winter wheat during the 2014 and 2015 growing seasons in Beijing, China. They discovered leaf area index overrides other climatic and biological factors such as WUE and crop yield, acting as a dominating role in the variability of seasonal T/ET. The seasonal variability in T/ET could be effectively explained via a power-law function of the LAI. It is also

found that to conserve water, the irrigation wetting layer should be controlled at a depth of 70 cm.

Comments: 1. Lacking long term observational water vapor isotope, the authors choose to partitioning ET using mass balance model. It is no problem in common sense. However, as shown in L248-L251, $\delta$E is assumed to be isotopic equilibrium with the soil water. This is not correct and it is the fatal flaw of this study. Both kinetic fractionation and equilibrium fractionation should be considered in this calculation. In addition, the value of $\delta$T is determined using the measured $\delta$ of stem water is also questionable. Many studies have shown that it is important to include non-steady state in $\delta$T estimation. These perhaps mean that the results should be considered in a qualitative sense and not trusted fully.

2. There is no any validation can be found in this study. Additional validations are required for mass balance model results. At least, simple model such as two-source model should be applied for comparing isotope method (e.g. Wang et al., 2015; Iso-SPAC), if direct measurement is not available. At the same time, how the author gets ET data? based on Eq 1? Is there any eddy-correlation measurement available?

3. Moreover, the sensitivity analysis should also be conducted, since T/ET is very sensitive to the bias of $\delta$T and $\delta$E.

4. About root water uptake depth, e.g. L294-L295: detailed result of ANOVA analysis is required. I think time series plots of isotopic compositions shown in Fig 1 are important and should be presented in somewhere of the manuscript.

By these reasons, I'd recommend rejecting this manuscript and give them plenty time for improving this manuscript and resubmission.